# Association of Androgenic Regulation and MicroRNAs in Acinar Adenocarcinoma of Prostate

**DOI:** 10.3390/genes13040622

**Published:** 2022-03-30

**Authors:** Julio Guilherme Balieiro Bernardes, Marianne Rodrigues Fernandes, Juliana Carla Gomes Rodrigues, Lui Wallacy Morikawa Souza Vinagre, Lucas Favacho Pastana, Elizabeth Ayres Fragoso Dobbin, Jéssyca Amanda Gomes Medeiros, Leonidas Braga Dias Junior, Gabriel Monteiro Bernardes, Izabel Maria Monteiro Bernardes, Ney Pereira Carneiro Dos Santos, Samia Demachki, Rommel Mario Rodriguez Burbano

**Affiliations:** 1Instituto de Ciências da Saúde, Universidade Federal do Pará, Belém 66050-160, Brazil; juliogbbernardes@gmail.com (J.G.B.B.); leonidas@supridad.com.br (L.B.D.J.); izabelmbernardes@gmail.com (I.M.M.B.); 2Núcleo de Pesquisas em Oncologia, Universidade Federal do Pará, Belém 66073-005, Brazil; julianacgrodrigues@gmail.com (J.C.G.R.); luivinagre@gmail.com (L.W.M.S.V.); lucas.favacho.celular@gmail.com (L.F.P.); elizabethdobbin7@gmail.com (E.A.F.D.); jessycamandag@gmail.com (J.A.G.M.); npcsantos@yahoo.com.br (N.P.C.D.S.); demachki@gmail.com (S.D.); rommelburbano@gmail.com (R.M.R.B.); 3Hospital Ophir Loyola, Belém 66063-240, Brazil; 4Faculdade de Medicina, Centro Universitário do Estado do Pará, Belém 66613-903, Brazil; bernardesgabriel2906@gmail.com

**Keywords:** nodular hyperplasia of the prostate, acinar carcinoma, androgen receptor, microRNA

## Abstract

Background: Prostate cancer represents 3.8% of cancer deaths worldwide. For most prostate cancer cells to grow, androgens need to bind to a cellular protein called the androgen receptor (AR). This study aims to demonstrate the expression of five microRNAs (miRs) and its influence on the AR formation in patients from the northern region of Brazil. Material and Methods: Eighty-four tissue samples were investigated, including nodular prostatic hyperplasia (NPH) and acinar prostatic adenocarcinoma (CaP). Five miRs (27a-3p, 124, 130a, 488-3p, and 506) were quantified using the TaqMan^®^ Real Time PCR method and AR was measured using Western blotting. Results: Levels of miRs 124, 130a, 488-3p, and 506 were higher in NPH samples. Conversely, in the CaP cases, higher levels of miR 27a-3p and AR were observed. Conclusion: In the future, these microRNAs may be tested as markers of CaP at the serum level. The relative expression of AR was 20% higher in patients with prostate cancer, which suggests its potential as a biomarker for prostate malignancy.

## 1. Introduction

Prostate cancer (CaP) is the fourth most common malignancy worldwide, and represents one of ten neoplasms with a higher mortality in men of all ages; as such, is considered a substantial public health problem [1,2].

The level of prostate-specific antigen (PSA) is the main screening method for CaP, often used in association with the digital examination of the prostatic gland [3]. Ultrasound-guided transrectal prostate biopsy is the diagnostic method for CaP, whose histopathological study can define tumor aggressiveness and disease prognosis. The risk stratification described by D’Amico [4] considers parameters such as clinical stage, PSA level, and Gleason score. 

CaP is a very heterogeneous tumor, limiting the effectiveness of parameters for diagnosis and risk stratification, which are currently based on clinical characteristics [3]. This limitation makes it difficult to distinguish between potentially lethal forms of the neoplasia and indolent forms of the disease, generating controversial overtreatment in cases of low-significance disease [5,6]. In this scenario, studies have attempted to identify effective biomarkers that can predict tumor aggressiveness and disease prognosis, especially those that can indicate tumor recurrence, such as castration-resistant prostate cancer [7].

For most prostate cancer cells to grow, androgens need to bind to a cellular protein called the androgen receptor (AR). Androgen receptor antagonists are drugs that connect to these receptors, preventing androgens from causing tumor growth; they are administered orally daily, and include flutamide, bicalutamide, and nilutamide. Blocking the androgen receptor with its respective androgenic ligand is an important basis in the treatment of acinar prostatic adenocarcinoma (CaP) [8].

The microRNAs (miRNAs or miRs) are responsible for the regulation of more than 60% of the human genome and are involved in many cellular mechanisms, such as proliferation, differentiation, migration, and apoptosis. Several studies have already demonstrated their involvement in all cancer hallmarks, and they have already demonstrated either oncogenic or tumor suppression functions [9]. 

miRNAs are interesting biomarkers used for diagnosis, prognosis, and molecular classification that may guide therapy for prostate cancer. This is due to their stability, the possibility of analysis in several body fluids, and, mainly, because they allow for differentiation between benign and malignant disease, or even to differentiate between an indolent and an aggressive type of disease [10].

In CaP, it has been shown that miRs can play an important role in regulating the expression of androgen receptors (ARs), affecting their expression and altering their functionality mainly through interference with the signaling and proliferation of androgenic cells [11,12]. In addition to the regulation of these receptors, miRNAs also demonstrate their importance in the diagnosis, prognosis, and therapeutic response of patients, showing a potential role as a biomarker for these three points [1].

AR is a binding-dependent transcription factor whose activation by androgens induces the transcription of target genes, resulting in the proliferation of CaP cells [13]. Studies have shown the association between several miRs and androgenic regulation. Among these, the miR-27a can act as an oncomiR, inducing increased growth of prostate cancer cells through repression of tumor suppressors, AR control, and prohibitin [14,15]. Prohibitin, in addition to acting as a repressor, is also a target protein for AR, and its negative regulation promotes cell growth, which demonstrate its tumor suppressor function [16]. The miR-130a has been reported to be involved in important oncogenic signaling pathways in prostate carcinoma, and it is negatively regulated in cancer tissue [17]. 

It has been demonstrated that the miR-124 acts as a tumor suppressor through negative regulation of the androgen receptor, inducing positive regulation in the p53 protein. Another reported tumor suppressor is the miR-488, which negatively regulates the transcriptional activity of AR, blocking proliferation and increasing apoptosis in CaP cells [11,12].

In addition, the miR-506 is involved in the epithelial–mesenchymal transition process, with a gain of metastatic potential, particularly in breast neoplasms [18,19].

Therefore, given the effect that various miRs exert on a variety of cellular and molecular pathways involved in the pathogenesis of prostate cancer, it seems likely that miR expression profiles can be considered potential candidates to prognosis, diagnosis, and/or treatment of this disease.

The miscegenation of the Brazilian population is a relevant factor in genetic studies, since the frequency of several polymorphisms varies between different geographical populations [20]. Such heterogeneity highlights the importance of understanding the distribution of clinically active variants in our population, which could help to provide a better clinical conduct of different diseases [21].

This investigation aimed to clarify the association of the relative expression of five miRs (27a, 124, 130a, 488, and 506) and AR as possible tumor biomarkers in patients with CaP and nodular prostatic hyperplasia (NPH) from the northern region of Brazil.

## 2. Materials and Methods

### 2.1. Tissue Samples

This investigation is classified as an analytical retrospective study. The miRNA and AR expression patterns were evaluated in 41 samples of prostate benign hyperplasia from patients undergoing prostate transurethral resection (TURP) and 43 prostate cancer tissue samples were obtained from patients undergoing radical prostatectomy in Northern Brazil.

All the patients had negative histories of exposure to either chemotherapy or radiotherapy before surgery, and there was no co-occurrence of other diagnosed cancers. Informed consent with approval of the Ethics Committee of the João de Barros Barreto University Hospital (Protocol #406007) was obtained from all patients prior to specimen collection.

Part of each tumor sample and benign hyperplasia was dissected and formalin-fixed and paraffin-embedded (FFPE). Sections of FFPE tissue were stained with hematoxylin-eosin for histological evaluation or used for immunohistochemistry (IHC) analysis. The clinical pathologic characteristics included age, American Joint Committee on Cancer (AJCC) stage, Gleason score, and PSA level. All the samples were classified according to the D’amico score, recently modified [22,23]. 

### 2.2. Western Blotting

Total protein was isolated from prostate tissue samples using the All Prep DNA/RNA/Protein Kit (Qiagen, Hilden, Germany) according to the manufacturer’s instructions. The protein pellet was dissolved in buffer containing 7 M urea, 2 M thiourea, 4% CHAPS, 50 mM DTT, 1% Protease Inhibitor Cocktail (Sigma–Aldrich^®^ Merck KGaA, Darmstadt, Germany), and 0.5% of each Phosphatase Inhibitor Cocktail 1 and 2 (Sigma–Aldrich^®^ Merck KGaA, Germany), as previously described by our group [24]. Protein concentration was determined by the Bradford method (Sigma–Aldrich^®^ Merck KGaA, Germany). Reduced protein (25 μg) from each sample was separated by a 12.5% homogeneous sodium dodecyl sulfate-polyacrylamide gel electrophoresis (SDS-PAGE) and electroblotted onto a polyvinylidene fluoride (PVDF) membrane Amersham^™^ Hybond^®^ P (Sigma–Aldrich^®^ Merck KGaA, Germany). The PVDF membrane was blocked with phosphate-buffered saline containing 0.1% Tween 20 and 5% low-fat milk and incubated overnight at 4 °C with the following primary antibodies: anti-AR primary antibody (dilution 1:10; MS-443-P1ABX; Thermo Fisher Scientific, Waltham, MA, USA) and anti-ACTB antibody (dilution 1:250; Ac-15; Thermo Fisher Scientific, USA). After extensive washing, a peroxidase-conjugated secondary antibody was added for 1 h at room temperature. Immunoreactive bands were visualized using the Western blotting Luminol reagent, and the images were acquired using an Image Quant 350 digital image system (GE Healthcare, Stockholm, Sweden). β-actin (ACTB) was used as a loading reference control. Western blot analysis was performed as previously described by Leal et al. (2014) [24]. The relative quantification of AR expression was the result of the comparison between the amount of AR in a sample of a patient with CaP and the amount of AR in the control sample, which is composed of a pool of 41 nodular prostatic hyperplasia (NPH). This pool works as a calibrator, therefore when the result of the comparison between the Cap sample and the pool is equal to 1, it means that the amount of AR in the patient sample with Cap and in the pool (control) is equal. Thus, when the result of the comparison between the tumor sample and the control is equal to 1.2, it means that there is 20% more AR in the tumor sample than in the control.

We used the pool as a tool to measure the relative quantification of CaP samples, both miRNA and protein, since normal tissue is rarely collected as biopsy samples in the routine of patients with CaP.

### 2.3. Relative Quantification of MicroRNA Expression

Total RNA (10 ng) was extracted with TRI Reagent^®^ Solution (Life Technologies, Carlsbad, CA, USA) following the manufacturer’s instructions. RNA concentration and quality were determined using a Nanodrop spectrophotometer (Thermo Scientific, Wilmington, DE, USA) and run on a 1% agarose gel. Reverse transcription was performed using a TaqMan Small RNA Assays kit (Applied Biosystems, Foster City, CA, USA) according to the manufacturer’s recommendations. The obtained complementary DNA (cDNA) was stored at −20 °C and later added to 96-well plates with 7.5 μL of 1x TaqMan Universal PCR Master Mix II (Life Technologies, Carlsbad, CA, USA) and 0.5 μL of TaqMan^®^ Advanced miRNA Assays for hsa-miR-27a-3p (477998_mir ASSAY ID), hsa-miRNA-124 (002197 ASSAY ID), hsa-miRNA-130a (462691_mat ASSAY ID), hsa-miR-488-3p (002357 ASSAY ID), and hsa-miR-506 (001050 ASSAY ID); (TaqMan MGB probe, and forward and reverse primers from Life Technologies, Carlsbad, CA, USA), according to the manufacturer’s recommendations. The SNORD7 assay (Life Technologies, Carlsbad, CA, USA) was selected as an internal control for monitoring RNA input and reverse transcription efficiency. We used the small nucleolar RNA SNORD7 because it is commercially distributed as a reference and recommended as a generally stable reference gene [25]. qRT-PCR was performed in an ABI Prism 7500 thermal cycler. All real-time PCRs were performed in triplicate, and the average of the results was used in subsequent analyses. The relative quantification (RQ) of gene expression was calculated according to Schmittgen and Livak [26]. In tissue sample analyses, a pool of miRNAs extracted from 41 NPH tissues was designated as a calibrator from each tumor [27]. When the relative quantification value is equal to 1, it means that the amount of miRNA is equal both in the tumor and in the pool of miRNAs extracted from 41 NPH. When the value is 1.5, it means that the relative quantification of that miRNA is 50% higher in the tumor than in the pool of miRNAs extracted from 41 NPH. Following this reasoning, when the relative quantification is equal to 0.5, it means that there is 50% less of this miRNA in the tumor than in the pool of miRNAs extracted from the 41 NPH samples. This same correlation was performed by Western blot using the pool of AR protein extracted from the 41 NPH samples.

### 2.4. Statistical Analyses

The analyses and graphs were created using the software R Studio v1.2.1355 associated with the R v.3.6.1 language. Mann-Whitney tests were used to analyze differences in quantitative variables between the groups investigated and in risk stratification. The Chi-square test (χ^2^) was used in the analysis of nominal values. The correlations between quantitative variables for the calculation of the correlation coefficient (r) and determination coefficient (r^2^) were made by the Pearson correlation test, in addition to a linear regression association for the formation of a trend line between the analyzed variables.

## 3. Results

Forty-three patients with prostate acinar adenocarcinoma that underwent radical prostatectomy comprised the case group, and 41 patients with nodular prostatic hyperplasia (NPH) that underwent prostatic transurethral resection comprised the study control group.

Patients with CaP were stratified according to the risk score for D’Amico tumor recurrence (high, intermediate, and low). The clinical data for this stratification method is shown in Appendix A. Patients with CaP and clinical staging between T1 and T2c who underwent curative prostatectomy were considered. Cases of transurethral resection with CaP (advanced) in this study were classified as high risk. 

The classification by D’Amico risk criteria for CaP showed that the 20% of the group with lower risk (n = 14) had a mean age of 72.2, PSA of 5.7, Gleasonscore ≤ 6. The intermediate risk (n = 24) group mean age was 66 years, PSA was 10 years, and Gleasonscore was 7, corresponding to 20% of the intermediate classification. The high-risk group (n = 5) averaged 59.6 (years), with a mean PSA of 131.4 ng/mL, and a predominant clinical staging in T2c.

The relative prostatic levels of the AR (Appendix A) and five miRs (27a-3p, 124, 130a, 488-3p, and 506) were measured in the CaP and NPH groups (Figure 1). Appendix A shows the median and variance of each miR expression levels in both groups, besides the *p* values obtained by the Mann-Whitney test. All miRs presented significant differential expression levels between the CaP and NPH groups. The levels of the miRs 124, 130a, 488-3p, and 506 were higher in the NPH group, similar to the action of a tumor suppressor RNA function. Conversely, the levels of the miR-27a-3p and AR were higher in CaP, suggesting an RNA with an oncogenic role.

The ROC curves for the miR-27a-3p, miRNA-124, miRNA130a, miR-488-3p, and miR-506 showed significant results (*p* < 0.001) (Figure 2).

The miR-27a-3p, miR-124, miR-488-3p, and miR-506 had an area under the ROC curve (AUC) equal to 1.00 (95% CI 1.0–1.0). The miR130a had an AUC equal to 0.999 (95% CI 0.998–1.00).

The mir-27a-3p was able to distinguish between the case and control groups when the result was greater than or equal to 1.217 (Sensitivity = 100%; Specificity = 100%). The miR-124, miR130a, miR-488-3p, and miR-506 showed an opposite mathematical behavior to that of miR -27a-3p, considering positive tests when the values were less than or equal to the cutoff point.

The cutoff points were 0.918, 0.798, 0.924, and 0.758, respectively. The miR-124, miR-488-3p, and miR-506 showed 100% sensitivity and 100% specificity for their cutoff points, while the miR130a showed 97.7% sensitivity and 100% specificity. All results referring to the ROC curve are described in Appendix A. We performed a statistical analysis to confirm the data shown in the ROC curve. All *p* values in the comparisons were less than 0.001. These data can be better visualized in the Appendix A.

After the prostatic expression levels of the five miRs and AR were determined in the CaP patients, it was possible to perform an association analysis of these levels with the D’Amico risk score stratification (Table 1); the mean and variance for each expression investigated was described. Among the differentially expressed miRs, the miR-488-3p and miR-506 were statistically significant.

Interesting results were obtained when comparing the prostatic levels of miR-488-3p between the low- and intermediate-risk groups, with mean values of 0.75 and 0.67, respectively (*p* value = 0.007). Figure 3 shows that the prostate levels of miR-488-3p were inversely proportional to the risk of tumor recurrence. The correlation suggests that the lower the risk score, the greater the expression of the miR-488-3p. Thus, this association is correlated with a low risk of tumor recurrence (intermediate vs. low; *p* value: 0.007).

A similar correlation was demonstrated for miR-506 (Figure 4), in which a significant association of patients with lower prostatic RNA levels in correlation with a higher risk of tumor recurrence was observed (high vs. low; *p* value: 0.047; high vs. intermediate; *p* value: 0.018). 

Figure 5 shows the differential relative expression of AR between CaP (red) and NPH (blue) patients, demonstrating that the cutoff point of the relative expression was 1.2, representing an increase of 20%, which allows to differentiate cases of benign hyperplasia from those with a malignant tumor. The relative expression value of AR is therefore demonstrated in this study as an important marker of malignant prostate neoplasms.

Additionally, it was possible to observe that, among the evaluated miRs, the miR-27a-3p presented a statistically significant moderate direct correlation with preoperative PSA levels (Figure 6) in patients with CaP (r: 0.321; *p* value: 0.038). 

## 4. Discussion

Knowledge about the molecular bases in the etiopathogenesis of NPH and CaP regarding their differences and possible similarities is still limited. CaP presents a heterogeneous genotype, often sharing signaling pathways between both NPH and CaP [28]. miRs are small non-coding RNAs that regulate gene expression by binding to target mRNA and are commonly deregulated in CaP, being partially responsible for the aggressive manifestation of the disease [29,30]. 

A slight increase in the critical levels of AR mRNA and its corresponding protein is sufficient to develop resistance to anti-androgenic therapy. The strategy to downregulate AR mRNA expression in combination with anti-androgenic therapy may prevent or delay the development for the hormone-independent condition in CaP [31]. In this sense, the present study demonstrated the performance of miRNAs with a regulatory function in the formation of AR. The hyperexpression of miRNAs that negatively regulate the transcriptional activity of the androgen receptor may inhibit the production of endogenous AR protein in CaP [32]. For this reason, the present work tested its miRNA expressions in both CaP and NPH samples.

In this study, the relative prostatic levels of the androgen receptor and five miRs (27a-3p, 124, 130a, 488-3p, and 506) were measured in individuals diagnosed with CaP or NPH. The growth of prostate tumors is dependent on circulating androgens that activate androgen receptors. Changes in mRNA splicing lead to constitutively active AR forms [12].

All the miRs analyzed presented significant expression levels between the CaP and NPH groups. The levels of miR-124, 130a, 488-3p, and 506 were higher in individuals with NPH, acting as tumor suppressor miRs.

Additionally, the concentrations of microRNAs obtained through Western blot analysis were evaluated for their performance in discriminating tissues from prostate cancer and prostatic hyperplasia. All five microRNAs evaluated in the present study showed significant results for different cutoff points of the ROC curve with significantly high rates of sensitivity and specificity, which demonstrates their potential as possible noninvasive diagnostic biomarkers for prostate cancer.

The mir-27a-3p was able to distinguish cases of prostate cancer with a cutoff point equal to or greater than 1.217, with maximum values of sensitivity and specificity (100%). This microRNA has also been associated in the literature in studies differentiating pancreatic cancer from benign pancreatic/peripancreatic diseases, with a sensitivity of 82.2% and specificity of 76.7% (AUC = 0.840; 95% CI, 0.787–0.885%) [33]. Moreover, the mir-27a-3p was shown to be able to identify prostate cancer metastasis (AUC of 89.5% (95% CI = 79.5–99.5%)) [34].

The miR-124, miR130a, miR-488-3p, and miR-506 were also capable of identify tissues with prostate cancer, but with cutoff points lower than 1 (0.918, 0.798, 0.924 and 0.758, respectively). All microRNAS showed sensitivity and specificity of 100%, except the miR130a, which had a sensitivity of 97.7%. This is the first study to evaluate these microRNAs with an ROC curve analysis to show their potential in diagnosing prostate cancer tissues.

Several studies have indicated that increased levels of miR-124 are related to the suppression of CaP through several pathways. Elevated levels of miR-124 inhibit invasion and proliferation of carcinogenic prostatic cells by mediating a transcription-activating pathway [35,36]. The miR-124 also targets AR transcription, acting as a tumor suppressor that widely limits prostate cancer growth [37,38]. On the other hand, lower levels of this miRNA are observed in prostate cancer, ensuring resistance to neoplasms.

Decreased regulation in the miR-130a is associated with the promotion of hypermethylation, leading to the inhibition of a malignant phenotype of prostate cancer [39]. Repression of AR signaling and mitogen-activated protein kinases are also associated to this decreased regulation, inducing cell cycle arrest and tumor cell apoptosis [17], in addition to being related to drug resistance [40,41].

Few studies have correlated the miR-488-3p and miR-506 with CaP or NPH. miR-488 is a miRNA that acts as a tumor suppressor in various tumors, such as stomach [42], kidney [43], liver [44], and skin cancer [45]. Positive regulation of the miR-488 in these tumors may result in the inhibition of gene expression. Wang et al. (2019) demonstrated that the miR-488 was not expressed in CaP and decreased the proliferation of neoplasms cells [46]. The miR-506 is negatively regulated in various types of human cancer. The expression of this miRNA was also reduced in CaP cells compared to normal prostatic epithelial cells, suggesting that it may also function as a tumor suppressor [47].

In this study, the miR-27a associated with CaP behaved similar to others in the literature, with oncomiR properties [30,48], so that it was overexpressed in patients with CaP when compared to the NPH group.

Our results suggest a directly proportional statistical correlation between the miR-27a-3p and PSA, a factor involved in the assessment of the risk of recurrence of CaP. Moreover, in this study, miRNA-27a-3p presented positive feedback with the expression of androgen receptors.

The miR-27a-3p targets the 3’UTR region of the prohibitin gene (PHB). PHB is an inhibitor of androgen receptor activity, therefore the reduction in PHB expression results in increased AR activity and the proliferation of prostatic cells. In addition, AR has been shown to be a post-transcriptional mediator of miR-27a-3p maturation [49]. That is, in prostate cancer, hyperexpression of miR-27a-3p reduces PHB activity, enabling greater expression of androgen receptors.

The expression of miR-27a is also related to increased cell proliferation, migration, and cell invasion through the activation of RAS and MAPK pathways [50], increase in the expression of angiogenic factors such as VEGF and Inos [51], reduction in BTG-2 activity, an antiproliferative and antiapoptotic protein [30,35,52], decrease in expression levels of E-cadherin, and various other proapoptotic molecules, such as Bcl-2, BAX, APAF1, and FOXO [35,46,52].

It is worth mentioning that, in some studies, the miR-27a behaves as a tumor suppressor so that overexpression inhibits cell proliferation and promotes cellular apoptosis in prostate cancer, which is contrary to the findings of this study [53,54].

The development of prostate cancer is dependent on androgens. AR is expressed in both epithelial and stromal cells and regulates cell differentiation and proliferation. In malignant cells, the target AR gene group is known to be involved in the onset and progression of cancer [55,56].

Castration-resistant prostate tumors also depend directly on AR [57], which can play a significant role in the transition from early to advanced disease and, consequently, is a valuable therapeutic target. Thus, it can be affirmed that CaP is directed by androgens, since AR is one of the most important key factors of cell growth in castration-resistant tumors [49].

Because of this, many current studies have researched several molecular markers that may interfere with the expression or signaling of AR in patients with CaP [48]. Moreover, the PSA test indirectly monitors tumor activity through AR signaling, which usually correlates with tumor load [58].

Finally, an association between staging and Gleason grade was determined in the current series. Most tumors were classified as T2b and T2c (Appendix A), with the most common histological grade being a Gleason score of 7 or higher. This association in most sample cases of a high Gleason score for a T2 staging was previously determined in the Brazilian population [59,60], therefore a large-scale study would be needed to clarify this phenomenon.

## 5. Conclusions

Our results showed that the overexpression of AR in CaP occurs in accordance with the deregulation of the expression of several microRNAs, which regulate the production of AR and present different quantifications of benign hyperplasia of the prostate. Moreover, we showed the promising role of miR-27a-3p, miRNA-124, miRNA130a, miR-488-3p, and miR-506 in distinguishing groups of prostate cancer and prostatic hyperplasia with rates of sensibility and specificity of 100%.

The relative expression of AR proved to be an important differential marker of prostate malignancy, since the results obtained from the relative expression of AR in patients with prostate cancer presented an expression above 20% (point of 1.2) compared to the patients with benign prostatic hyperplasia.

The collection of NPH samples for use in a pool is a tool that allows to measure CaP biopsies, since transition zones (PHN site) are rarely collected as biopsy samples in the routine of patients with CaP. One of the limitations of this study is the use of tissue expression of NPH as a control to measure the expression of CaP, since it is an invasive test, but in the future these miRNAs can be tested for their use as markers of CaP at the serum level.

In this study, the risk of recurrence according to the D’Amico score was inversely proportional to the level of the miRs 488 and 506. Therefore, the findings that the relative quantifications of AR and miR-27a-3p were reduced in NPH and elevated in CaP also presented a proportionally direct relationship with serum PSA levels.

## Figures and Tables

**Figure 1 genes-13-00622-f001:**
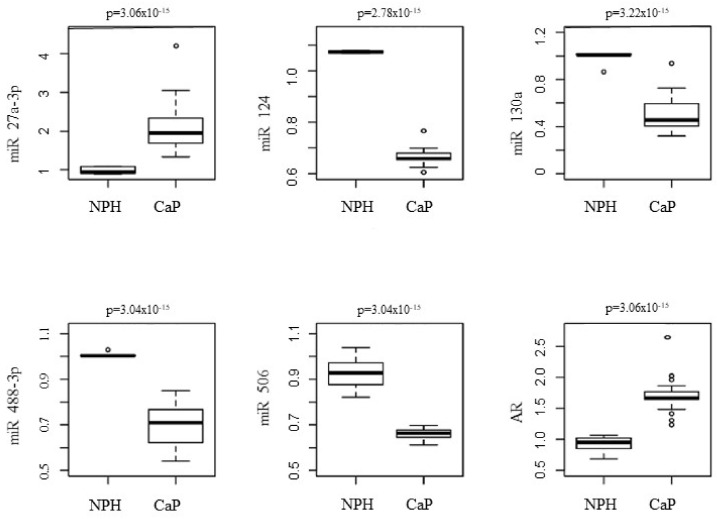
Correlation of the relative prostatic levels of the miRs 27a-3p, 124, 130a, 488-3p, and 506 between the CaP and NPH groups.

**Figure 2 genes-13-00622-f002:**
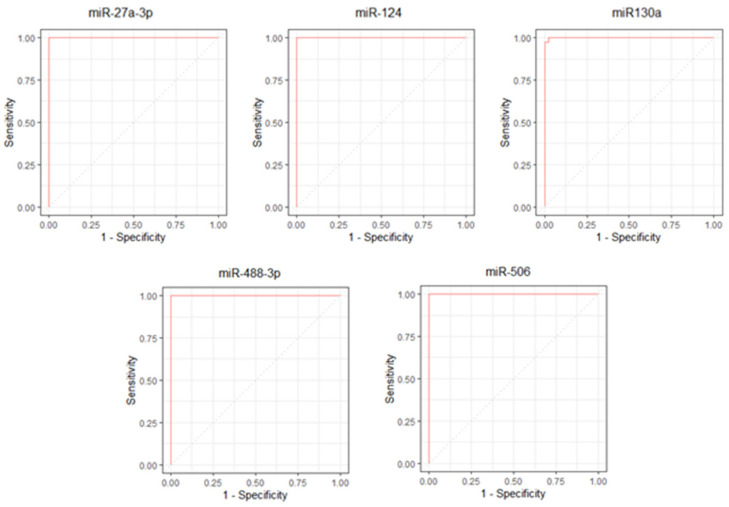
ROC curves for the miR-27a-3p, miR-124, miR-130a, miR-488-3p, and miR-506.

**Figure 3 genes-13-00622-f003:**
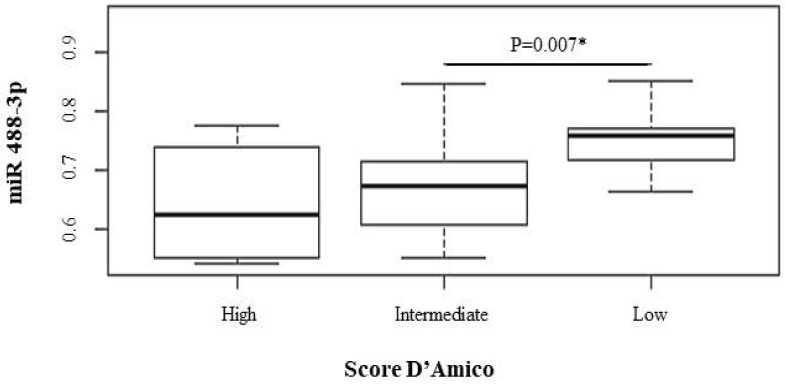
Distribution of Score D’Amico and relative prostate levels of miR488-3p in the CaP patient group. * significative *p* value (<0.05). All *p* values were determined using the Mann-Whitney test. The Y axis represents the expression of miRNA by the Western blotting method and the X axis represents the distribution of the sample from the case group according to risk stratification D’Amico Score.

**Figure 4 genes-13-00622-f004:**
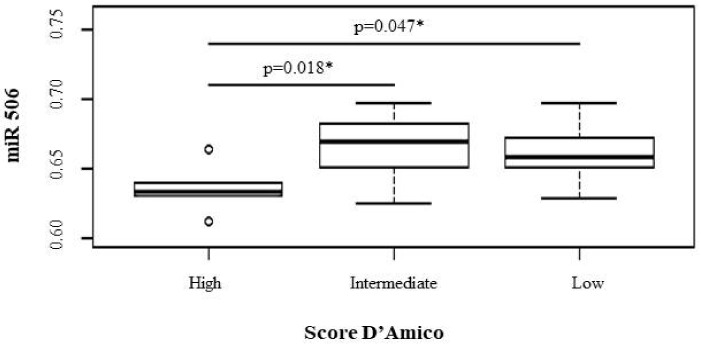
Distribution of Score D’Amico and relative prostatic levels of miR 506 in patients with CaP. * significative *p* value (<0.05). All *p* values were determined using the Mann-Whitney test. The Y axis represents the expression of miRNA by the Western blotting method and the X axis represents the distribution of the sample from the case group according to risk stratification D’Amico Score.

**Figure 5 genes-13-00622-f005:**
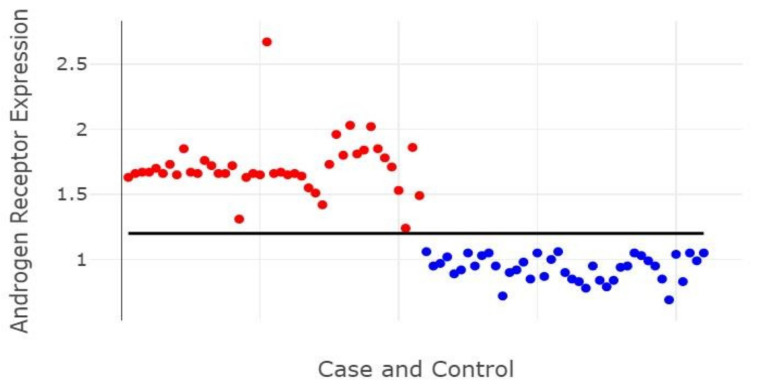
Correlation of the relative expression of the androgen receptor (AR) with NPH (blue) and CaP (red) patients. The *Y*-axis represents the expression of androgen receptor by the Western blotting method, and the *X*-axis represents each individual in the sample. Blue dots—control group; red dots—group case; black line—value of 1.2 expression of androgen receptor in the Western blot method.

**Figure 6 genes-13-00622-f006:**
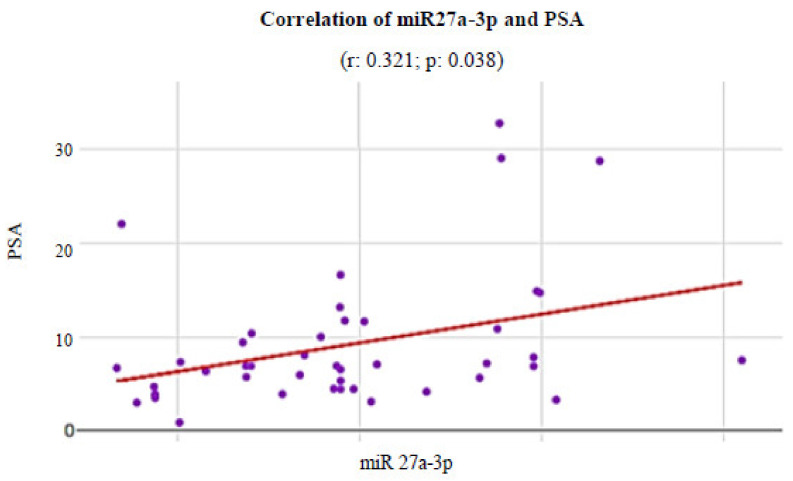
Correlation of PSA levels with the relative prostatic levels of miR27a-3p in the CaP patients: correlation coeficient; *p*: *p* value correlation. The Y-axis represents blood PSA levels and the X-axis represents the expression of miRNA by the Western blotting method.

**Table 1 genes-13-00622-t001:** Comparison by the D’Amico score and the relative prostate levels of AR and miRs investigated.

miR	Score D’Amico μ (h)			*p* Value *		
	High (n = 5)	Intermediate (n = 24)	Low (n = 14)	H-L ^a^	H-I ^b^	L-I ^c^
27a-3p	2.48 (1.08–3.87)	1.99 (1.87–2.10)	1.89 (1.51–2.27)	0.395	0.326	0.241
124	0.65 (0.57–0.74)	0.67 (0.65–0.68)	0.67 (0.63–0.69)	0.272	0.319	0.540
130a	0.45 (0.28–0.61)	0.52 (0.45–0.58)	0.45 (0.41–0.49)	0.864	0.386	0.455
488-3p	0.65 (0.51–0.77)	0.67 (0.63–0.70)	0.75 (0.71–0.78)	0.079	0.665	0.007
506	0.64 (0.61–0.66)	0.67 (0.65–0.67)	0.66 (0.64–0.67)	0.047	0.018	0.581
AR	1.81 (1.15–2.46)	1.69 (1.62–1.76)	1.66 (1.62–1.69)	0.777	0.862	0.558

* Mann-Whitney Test; μ (IC95%): Median expression and IC95%; ^a^ High-Low; ^b^ High-Intermediate; ^c^ Low-Intermediate.

## Data Availability

All relevant data will be shared as Appendix A files if the manuscript is accepted for publication.

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
