# Peer review of "Association of Androgenic Regulation and MicroRNAs in Acinar Adenocarcinoma of Prostate"

_genes, 2022, doi:10.3390/genes13040622_

Round 1

Reviewer 1 Report

Title: Association of Androgenic Regulation and MicroRNas in Acinar Adenocarcinoma of Prostate

This manuscript is well-designed and comprehensive, but I have some minor modifications needed to be clarified:

  1. in the title, the word MicroRNas should be replaced with MicroRNAs.
  2. Why the authors choose to evaluate the expression of these  five miRs (27a-3p, 124, 130a, 488-3p and 506) and what is the pathogenesis with prostate carcinoma?
  3. The whole manuscript needs to be improved to avoid grammatical errors.

Author Response

Question 1: In the title, the word MicroRNas should be replaced with MicroRNAs.

Response 1: Thanks for the suggestion and we changed the quote in the title

Question 2: Why the authors choose to evaluate the expression of these  five miRs (27a-3p, 124, 130a, 488-3p and 506) and what is the pathogenesis with prostate carcinoma?

Response 2: Many studies are being carried out in view of the analysis of miRNA expression profile, revealing them as important biomarkers of cancer predisposition and classification of different types of tumors, as well as in predicting therapeutic response, global prognosis and also, as identification of recurrences and metastasis. MiRNAs interfere as modulators of gene expression, through post-transcriptional regulation of expression. The choice of miRNAs was based on 1) scientific evidence observed in published studies available for consultation; 2) data obtained from the Gene Oncology database. We describe some of the references on which the choices of these markers were based:

  1. For mir-124, consideration was given to the description of its involvement in the negative regulation and tumor suppressive action, under the androgen receptor, and subsequently induction of the positive regulation of the p53 protein. The expression of hsa-miR-124 has been reported to be significantly reduced in malignant prostatic cells compared to cells with BPH (Shi et al., 2013).
  2. For mir-130a, its influence on two important oncogenic signaling pathways in prostate carcinoma, and downregulated in cancer tissue, was described. It is involved in the repression of gene expressions known to be overexpressed in cancer. It has been associated with action on targets of MAPK components, androgen receptor (AR) signaling, and androgen receptor co-regulators (HRAS), both signaling have a central role in the development and progression of prostate cancer (Boll et al. al., 2013). In the literature, its positive and negative regulation has been associated with several types of cancer, characterizing the complexity and diversity of hsa-miR-130a regulation in tumorigenesis (LIU et al., 2013).
  3. hsa-miR-27a can act by stimulating tumor development, by resembling oncogenic activity, in the regulation of the androgen receptor in prostate cancer. The increased expression of hsa-miR-27a causes the reduction of Prohibitin mRNA (PHB) and increases the expression of androgen receptor target genes with growth of prostate cancer cells. It is a miR-mediated androgen regulation, where hsa-miR-27a upregulates the androgen receptor, just as androgens regulate the expression of miR-27a transcriptionally and post-transcriptionally (FLETCHER et al., 2012).
  4. hsa-miR-506 has been reported to interact with the epithelial-mesenchymal transition (EMT). The suppressive role of miR-506 in TGFβ-mediated (EMT) induction has been documented, with the suppression of the expression of EMT-related genes, such as SNAI2, CD151 and CDH1, thus assuming tumor suppressive action in cancer progression. , by inhibiting the expression of genes that induce mesenchymal epithelium transformation (ARORA et al., 2013)
  5. 488-3p has been shown to be an important biomarker for predicting therapeutic response and relapse. Blockade of the androgen receptor with its respective androgen ligand constitutes an important basis in the treatment of PCa, therefore the strategy to down-regulate the expression of the Androgen Receptor mRNA in combination with antiandrogen therapy may prevent or delay the development to the hormone condition. -independent in the CaP. Studies have addressed the action of miR-488 with tumor suppression function in various cancers. Overexpression of miR-488 negatively regulates the transcriptional activity of the androgen receptor and inhibits the production of endogenous AR protein in the two conditions of CaP (androgen dependence and androgen independence). It was previously described that miR-488 blocks proliferation and enhances apoptosis in prostate cancer cells (SIKAND et al., 2011).

Question 3: The whole manuscript needs to be improved to avoid grammatical errors.

Response 3: Following the reviewer's recommendation, we performed a comprehensive grammar review of the article.

Reviewer 2 Report

Dear Editor The idea of manucript is an intersting one and manuscript is written well.

Author Response

We appreciate the considerations
